# Substrate-dependent fish have shifted less in distribution under climate change

Sarah M. Roberts [1]✉, Andre M. Boustany [2] & Patrick N. Halpin [1]

Analyses of the impacts of climate change on fish species have primarily considered dynamic oceanographic variables that are the output of predictive models, yet fish species distributions are determined by much more than just variables such as ocean temperature. Functionally diverse species are differentially influenced by oceanographic as well as physiographic variables such as bottom substrate, thereby influencing their ability to shift distributions. Here, we show that fish species distributions that are more associated with bottom substrate than other dynamic environmental variables have shifted significantly less over the last 30 years than species whose distributions are associated with bottom salinity. Correspondingly, species whose distributions are primarily determined by bottom temperature or ocean salinity have shifted their mean centroid and southern and northern range boundaries significantly more than species whose distributions are determined by substrate or depth. The influence of oceanographic versus static variables differs by species functional group, as benthic species distributions are more associated with substrate and they have shifted significantly less than pelagic species whose distributions are primarily associated with ocean temperatures. In conclusion, benthic fish, that are more influenced by substrate, may prove much less likely to shift distributions under future climate change.

[1] Marine Geospatial Ecology Lab, Nicholas School of the Environment and Earth Sciences, Duke University, Durham, NC 27708, USA. [2] Monterey Bay Aquarium, 886 Cannery Row, Monterey, CA 93940, USA. ✉email: sarah.m.roberts@duke.edu

Over the last several decades, anthropogenic greenhouse gas emissions have driven considerable increases in global and regional ocean temperatures and have transformed marine habitats[1]. These abiotic factors have, in turn, influenced marine fish at individual, population, and ecosystem-wide levels[2]. Historically, species have shifted distributions in response to climate change[3,4], but the explanations for differences in species' responses to climate change has largely been understudied. One reason for this is that research on the impacts of climate change on fish species has primarily considered dynamic oceanographic changes and has largely ignored the constraining role of other static physiographic variables on determining a species distribution. When examining both dynamic oceanographic and static physiographic variables, we can begin to understand the mechanisms that drive differential species shifts in distributions among functionally distinctive species groups.

Previous studies that have mainly considered dynamic oceanographic variables to reconstruct future marine fish distributions have generally found temperature to be a significant, if not the main driver in species distribution shifts[5]. This outsized focus on temperature is not surprising given the important role ocean temperatures play in individual species physiological constraints[6], the impact of temperature on other oceanographic variables such as productivity and oxygen[7], the strong changes in temperature over the last several decades especially in the Northeast Large Marine Ecosystem (LME)[8], and the robust temperature projections for the near future[9]. Yet, using temperature as the only proxy for a species' environmental niche may be inappropriate, especially in an oceanographically and physiographically dynamic region such as the North Atlantic[10]. Other factors such as benthic substrate[11], salinity[12] and depth[13] can play an important role in determining a fish species' distribution. We find that the historic ability of species to shift distributions varies depending on their life history type and resulting relationship with habitat-defining variables such as temperature, salinity, depth, and benthic substrate. By examining the differential effects of ecological variables on pelagic fish (i.e., species that inhabit the water column) versus benthic species (i.e., species that inhabit the seafloor) versus demersal or benthopelagic species (i.e. species that inhabit near the seafloor), we can begin to understand why certain species have shifted distributions over the past 30 years, while others have maintained a more static geographic distribution.

The US. Northeast LME provides an ideal case study to test the differences between pelagic, demersal and benthic fish species' historic shifts in geographic distributions in relation to the importance of oceanographic variables, substrate, and depth on their distributions. This region has seen greater changes in temperature than any other LME[8], has high resolution datasets on bathymetric depth and substrate, and has one of the longest scientifically collected time series of fish distribution and abundance. We used Generalized Additive Models (GAMs) to explain the impacts of oceanographic and physiographic variables on species distributions for the entire period between 1986 and 2018 for the spring and the fall. We then linked the strength of a species association with the environmental covariate (measured as deviance explained) to their historical shifts in distributions (biomass weighted centroids), shifts in range size (biomass weighted 95% kernel densities) and shifts in the northern and southern edge of a species range between the first 5 years (1986–1990) of the time period and the last 5 years (2014–2018) of the time period. We also linked species life history type (pelagic, demersal, or benthic; hereafter referred to as species type) to historic shifts in distributions and the species associations with environmental covariates (See Supplemental Data 1 and 2). Of the 93 species used in the fall, 13 were benthic, 63 were demersal, and 17 were pelagic and in the spring 12 were benthic, 63 were demersal and 16 were pelagic species (spring $n = 91$).

## Results and discussion

Our results show that a species' distribution—measured as change in mean center of biomass and change in northern and southern range extent—depends on the species relationship with oceanographic versus static environmental variables. In the fall, species whose distributions are most influenced by bottom salinity have shifted mean centroids of biomass significantly more than species whose distributions are most influenced by depth or substrate (nonparametric pairwise Wilcoxon test $p = 0.00007$ and $p = 0.0013$, Fig. 1a). Species whose distributions are most influenced by bottom temperature have shifted mean centroids of biomass significantly more than species whose distributions are most influenced by depth (nonparametric pairwise Wilcoxon test $p = 0.0083$, Fig. 1a) and shifted the northern and southern extent of their biomass weighted ranges significantly more than species whose distributions are most influenced by depth and substrate (nonparametric pairwise Wilcoxon test $p = 0.00018$ and $p = 0.014$, Fig. 1c and nonparametric pairwise Wilcoxon test $p = 0.0038$ and $p = 0.024$, Fig. 1d). Percentage change in biomass weighted range size did not depend on the strongest predictor variable for a species distribution (Fig. 1b). These patterns were not as strong in the spring (see Supplemental Fig. 1).

Our results show that in the spring and fall, pelagic species' distributions are primarily influenced by ocean temperature and depth, while demersal species' distributions are predominately influenced by ocean temperature and substrate, and benthic species' distributions are influenced by substrate (Fig. 2).

Pelagic species have shifted mean center of biomass significantly more over the historic time period compared with benthic species in the fall and significantly more than demersal species in the spring (nonparametric pairwise Wilcoxon test $p = 0.039$ and $p = 0.045$, Figs. 3a and 4a). Similarly, pelagic species have expanded their range size significantly more than demersal species and have shifted the northern extent of their range boundary significantly more than demersal species in the spring (nonparametric pairwise Wilcoxon test $p = 0.024$ and $p = 0.028$, Fig. 4c, d).

These results indicate that benthic species, more influenced by substrate than pelagic species, have retained their historical distributions in response to climate change, while pelagic species have shifted drastically. Exemplars of benthic fish that have retained their distributions include American plaice (*Hippoglossoides platessoides*), witch flounder (*Glyptocephalus cynoglossus*), and winter flounder (*Pseudopleuronectes americanus*) (Fig. 5a). These species distributions are strongly influenced by benthic substrate (Supplemental Data 1 and 2). Exemplars of pelagic fish that have shifted their distributions include rough scad (*Trachurus lathami*), Atlantic menhaden (*Brevoortia tyrannus*), and round herring (*Etrumeus teres*) (Fig. 5b). These species distributions are strongly influenced by bottom temperature and salinity (Supplemental Data 1 and 2).

By linking the historic evidence of species distribution shifts with their relationship with bottom temperature, bottom salinity, and benthic substrate, we have identified a broad generalization on how species with specific life history traits may be influenced by future climate change. Recent work suggests that pelagic species may shift farther under climate change compared with benthic invertebrates[14], and recent studies have included sediment type as a constraining variable in projections of future species distributions[15]. In conjunction with this work, our research highlights that these strong shifts have already occurred historically, and they are influenced by the strength of a pelagic

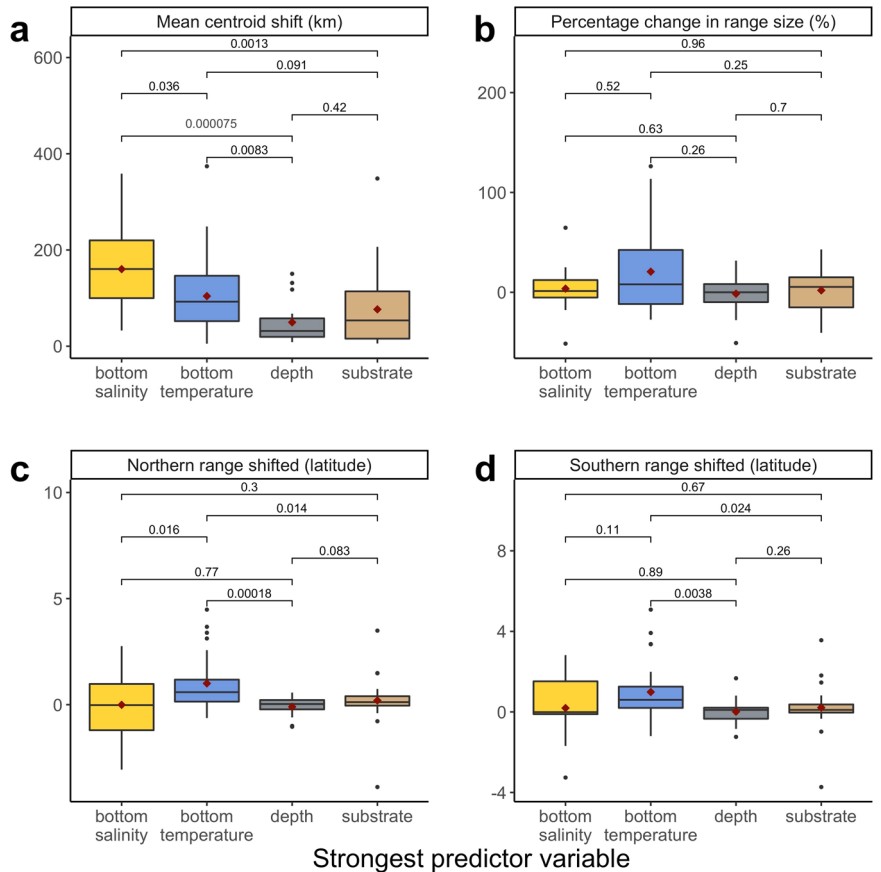

**Fig. 1 Historic distribution shifts versus the most important predictor variable for a species distribution in the fall.** Shifts in mean centroid of biomass from first 5 years to last 5 years in kilometers (**a**), percentage change in biomass range size from first 5 years to last 5 years in meters squared (**b**). Change in northern (**c**) and southern (**d**) extent of the biomass weighted range from first 5 years to last 5 years ($n = 93$). Brackets and numbers represent p-value. Whiskers represent 1.5* interquartile range. Box represents interquartile range as distance between first and third quartiles. Line represents median, red point represents mean, and black points represent outliers (outside of 1.5*IQR).

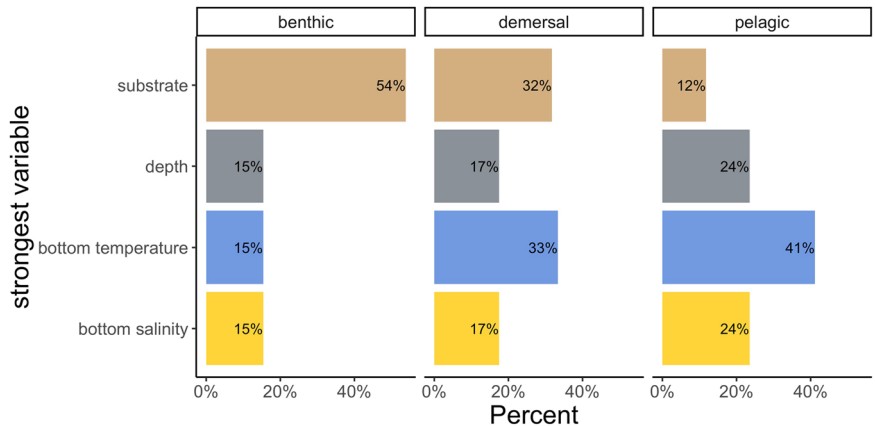

**Fig. 2 Strongest predictor variable for each species group in the fall.** Percentage of each species group that had substrate, depth, bottom temperature, and salinity as the strongest predictor variable in terms of deviance explained for the entire time series ($n = 93$). Results were similar across seasons (see Supplemental Fig. 2).

fish species' relationship with bottom temperature or salinity, compared with benthic species, which are more influenced by substrate. Given the importance of bottom substrate on benthic species distributions, we may see shifts in population dynamics, such as abundance and productivity as temperatures warm instead of geographic shifts in distributions. For, increased ocean warming that may go beyond their preferred thermal envelope is

expected in the Northeast LME and Mid Atlantic[9]. If affinity for substrate type keeps benthic species in regions that become too warm, these species may find themselves in suboptimal conditions and will not be able to relocate as easily as pelagic species.

Moreover, we examine demersal species that are influenced by both bottom temperature and substrate and have shifted moderately compared to benthic or pelagic species. Research in the

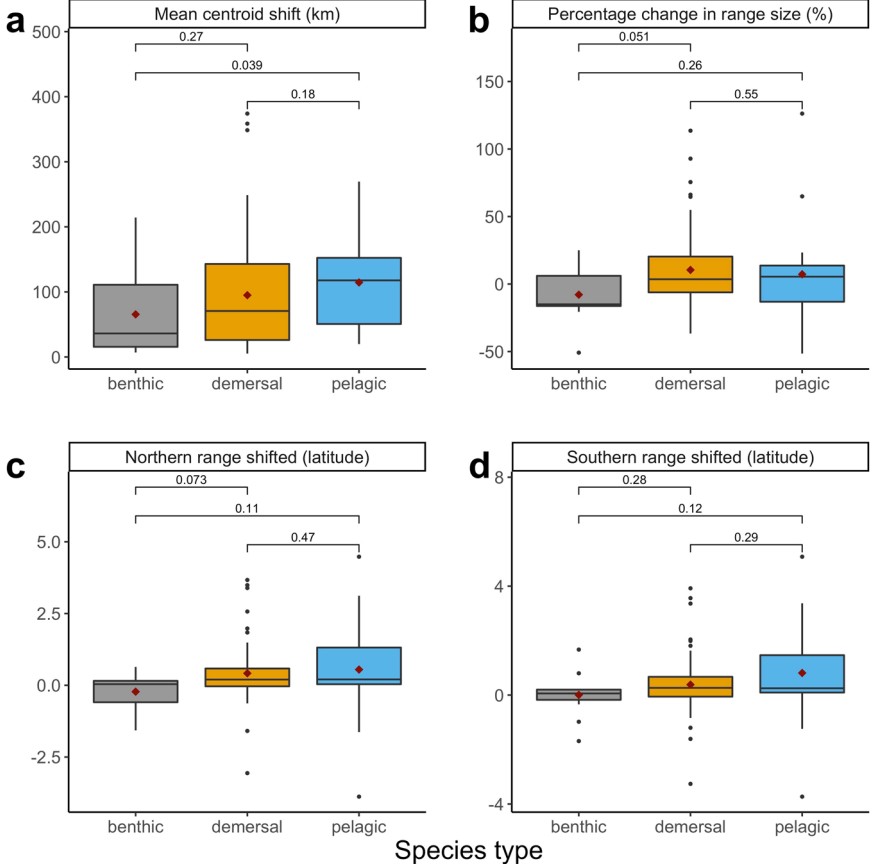

**Fig. 3 Historic distribution shifts for three species types in the fall.** Fall shifts in southern mean centroid (**a**), percentage change in range size (**b**), shifts in northern range boundary (**c**), and southern range boundary (**d**) from first 5 years and last 5 years in latitudinal degrees for each species group ($n = 93$). Brackets and numbers represent p-value. Whiskers represent 1.5* interquartile range. Box represents interquartile range as distance between first and third quartiles. Line represents median, red point represents mean, and black points represent outliers (outside of 1.5*IQR).

North Sea suggests that demersal species are shifting to deeper waters, which may suggest an interaction between bottom temperature and depth that requires further research[16]. Research in the Northeast LME has linked the shrinking spatial distribution of cusk, a demersal species, to a combination of ocean warming and the ensuing fragmentation of suitable bottom habitat[17], suggesting another interaction requiring future examination. As an intermediate case, these species may be the most unpredictable under climate change, and thus fisheries management will have to consider the varying nature of these species' distribution shifts.

These results provide historical evidence of pelagic species shifting distributions while benthic species remain more associated with their preferred substrate, which is most likely a result of the functional differences between these types of species. For example, the recruitment success of Atlantic menhaden (*Brevoortia tyrannus*), a pelagic schooling species we identified to shift historical distributions, is strongly related to ocean temperatures and larger climate dynamics such as the Atlantic Multidecadal Oscillation which influences temperatures and salinity[18]. The suitable habitat and migration timing of mackerel (*Scomber scombrus*), a pelagic schooling species, has been linked to changes in ocean temperatures and multidecadal variability[19], relying on temperature cues for their seasonal migrations. Research in the North Sea and Baltic Sea suggest that the northward shift of anchovies *(Engraulis encrasicolus)* and sardines *(Sardina pilchardus)*, two pelagic species, is strongly linked to temperature[20]. Benthic species, on the other hand, rely on structured biotic habitats, such as marshes, coral reefs, and submerged aquatic vegetation as well as abiotic sediment for their survival[21]. The

importance of abiotic sediment stems from the productivity of these habitats, as they usually contain high levels of detritus, microbes, and microinvertebrates[22].

It is important to identify the limitations of the approaches used in this study. The original species-CPUE data were collected from North Carolina to the Gulf of Maine, meaning we sampled a realized niche of the studied species. In comparison, the fundamental niche of the species may extend beyond our study area, in both the southern and northern directions. This study examined the role of bottom temperature, salinity, depth and substrate on species distributions, but was unable to examine the potential effects of fishing pressure, interspecific interactions, demographic changes, population sizes, or larval dispersal on species distributions and abundance[23]. While research has demonstrated that simple area-weighted center of distributions can be biased, we attempted to account for this by using several metrics of distribution (area occupied, northern, and southern extents of ranges)[24].

Despite these limitations, we expect our results will be valuable for fisheries managers as they anticipate the likelihood of species distribution shifts in their management areas. While current work has examined the role of temperature in determining species shifts, our work highlights the importance of examining the differential role of other static ecological variables on determining a species likelihood of shifting distributions under climate change. By uncovering the differential effects of certain habitat constraints on pelagic versus demersal and benthic species, we can begin to understand why certain species have shifted dramatically over the last thirty years, while others have retained their historical

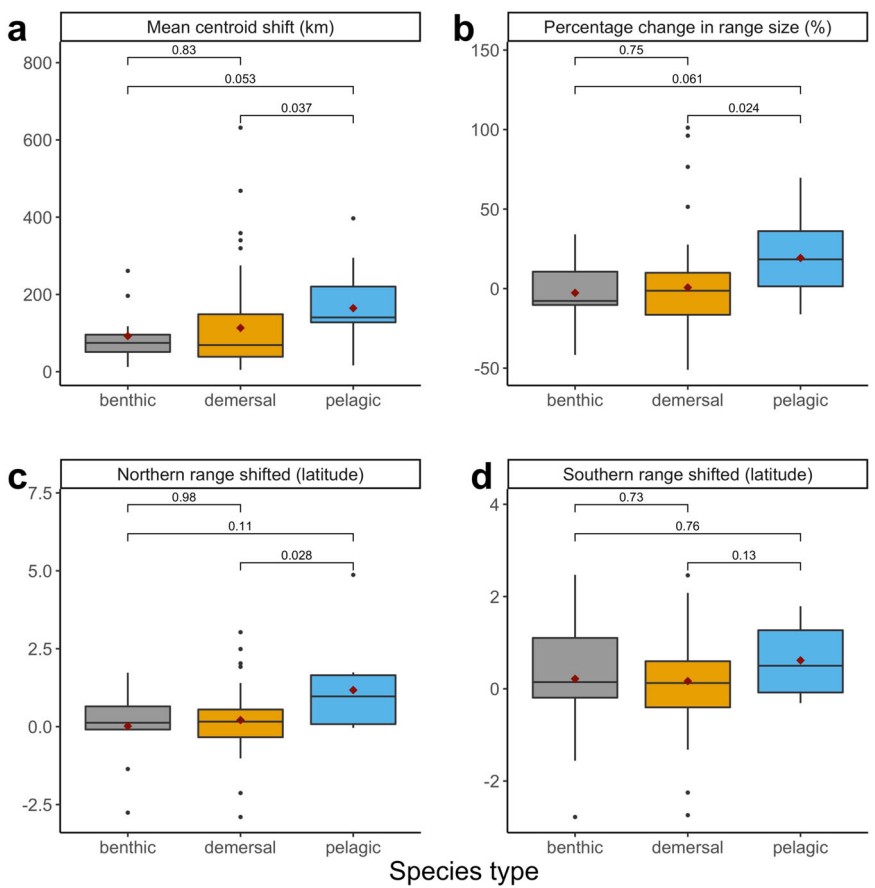

**Fig. 4 Historic distribution shifts for three species types in the spring.** Spring shifts in southern mean centroid (**a**), percentage change in range size (**b**), shifts in northern range boundary (**c**), and southern range boundary (**d**) from first 5 years and last 5 years in latitudinal degrees for each species group (*n* = 91). Brackets and numbers represent p-value. Whiskers represent 1.5* interquartile range. Box represents interquartile range as distance between first and third quartiles. Line represents median, red point represents mean, and black points represent outliers (outside of 1.5*IQR).

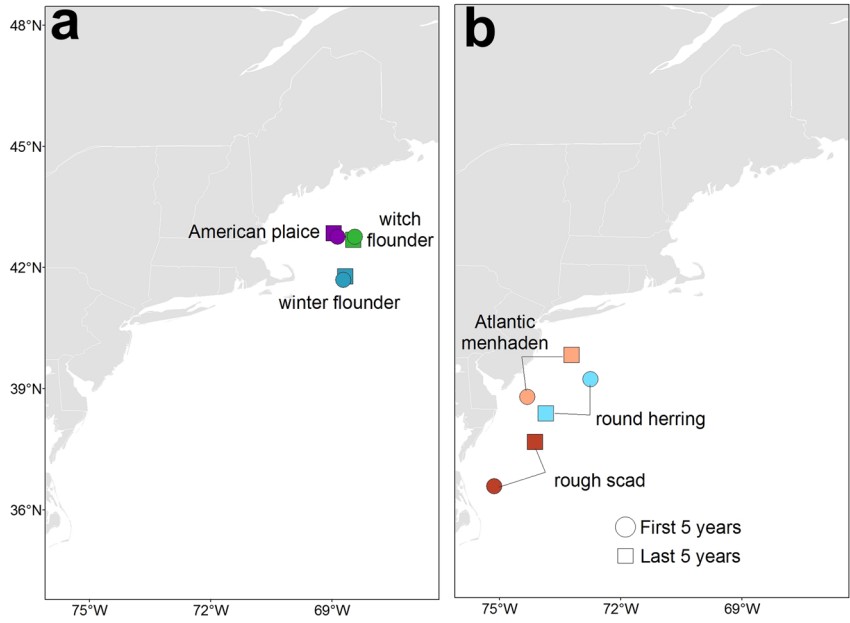

**Fig. 5 Changes in the biomass weighted mean centroid of species distributions in the Fall.** Biomass weighted mean centroids were calculated for two time periods: time period 1 (1986–1990) and time period 2 (2014–2018). Benthic species associated with bottom substrate retain their historical distributions (**a**) whereas pelagic species have shifted distributions (**b**). Species were selected to show extreme shifts and extreme retentions of distributions, and all shifts can be found in Supplemental Data 1 and 2.

distributions. These results highlight the need for stock assessments and future species distribution models that include the functional differences among species as well as the environmental variables that constrain their distributions in order to more appropriately understand the potential impacts of climate change on their future distributions. Only then can we understand how future fisheries will be impacted by the ecological effects of climate change.

## Methods

**Datasets.** Species catch per unit effort (CPUE) data were obtained from the National Oceanographic and Atmospheric Administration (NOAA) Northeast Fishery Science Center (NEFSC) U.S. NES bottom trawl survey, which has been conducted for almost 50 years in the spring and the fall and collected abundance and distribution data for over 250 fish species. The survey employs a stratified random design, with stations allocated proportionally to the stratum area. A 12 mm mesh cod liner is used to retain smaller bodied and juvenile fish, with all fish caught being weighed and counted[25]. We downloaded the data from OceanAdapt[4], which calibrates the CPUE for each species from the different survey ships used. We cleaned the data, excluding certain years as well as species that were not consistently sampled (excluded years prior to 1986 (data begin in 1968) due to irregular sampling of the southern strata, only included strata that were consistently sampled in the spring and fall (sampled each year from 1986–2018), and included 93 species in the fall and 91 species in the spring (Supplemental Fig. 3, Supplemental Data 1 and 2). Species were included if they were present in at least half of the years in both the spring and the fall (16 out of 33 years) and present in the first 5 years and the last five years in the fall and spring (>20 CPUE in surveys in 1986–1990 and 2014–2018). We only included fish species (bony fish and cartilaginous fish), as comparing fish species to invertebrates may be inappropriate. We grouped species that inhabit the seafloor as benthic, species that inhabit near the bottom as demersal, and species that inhabit the water column as pelagic (See Supplemental Data 1 and 2). Groupings were based on McHenry et al's study[14] and Fishbase classifications for additional species[26]. We compared McHenry et al's classifications to Fishbase and they were similar.

The study extent includes the Mid Atlantic Bight, Southern New England, The Gulf of Maine, and Georges Bank. Ocean temperature, salinity, and depth were collected in situ. Benthic substrate data were obtained from The Nature Conservancy's Northwest Atlantic Marine Ecoregional Assessment (grain size in mm)[27] (Supplemental Fig. 4). Annual, winter, and monthly North Atlantic Oscillation indexes were added based on the year collected, but we removed these variables from the final analysis as the deviance explained was minimal.

**Modeling.** We modeled the influence of environmental variables on species-CPUE using Generalized Additive Models (GAMs) with a negative binomial error distribution that had a log-link function, penalized regression splines, a REML smoothing parameter with an outer Newton optimizer, 10 knots, and omitted NAs. GAMs are a semiparametric extension of the generalized linear model (GLM) and are commonly applied to distribution and abundance studies for fishes[28]. GAMs utilize a smoothing function that can easily handle nonlinear relationships[29]. We calculated deviance explained by each predictor by running individual GAMs for each variable and species-CPUE combination for the entire time series (1986–2018) (See Supplemental Data 1 and 2) and recorded deviance explained versus the null model. We also determined the strongest predictor variable for each species as the predictor variable with the largest deviance explained. All GAMs were built using the mgcv package in RStudio[30].

To calculate shifts in species distributions over time we calculated the biomass weighted mean centroid of each species in two time periods: 1986–1990 and 2014–2018 using the spatial Eco package in R[31]. We calculated shifts in distributions as the geodesic distance between the two biomass weighted mean centroids for each species using the geosphere package in R[32]. We calculated changes in range size using the spatial kernel density function (weighted by CPUE, using gaussian kernels) in the spatial Eco package in R. We calculated each species range as the area with 95% of the populations kernel density for the two time periods as above. We calculated percentage change in range size as the present range size minus the past range size divided by the past range size. Changes in minimum and maximum latitude for each species were calculated using the 95% kernel density range from above. For species-specific geographic shifts see Supplemental Data 1 and 2.

**Statistics and reproducibility.** We performed two-sided Wilcoxon nonparametric tests to assess the significance of historical changes in species distributions (mean centroid, percentage change in range size, minimum and maximum latitude) between the strongest predictor variables (obtained from GAMs run on total time series) as well as between the three species groups. Of the 93 species used in the fall, 13 were benthic, 63 were demersal, and 17 were pelagic, and in the spring 12 were benthic, 63 were demersal and 16 were pelagic species (spring $n = 91$). For final datasets used to make the figures, see Supplemental Data 1 and 2.

**Reporting summary.** Further information on research design is available in the Nature Research Reporting Summary linked to this article.

## Data availability
All of the data analyzed in this study are publicly available. NEFSC bottom trawl data may be downloaded from OceanAdapt (https://oceanadapt.rutgers.edu). The substrate data can be downloaded from http://www.conservationgateway.org. Final datasets used to create figures can be downloaded from github (https://github.com/sr197/Sticky_Fish) or Zenodo (https://doi.org/10.5281/zenodo.4000171)[33].

## Code availability
All analyses were conducted in R version 3.5.2. Code used to run the analysis and create figures can be downloaded from github (https://github.com/sr197/Sticky_Fish) or Zenodo (https://doi.org/10.5281/zenodo.4000171)[33].

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

## Acknowledgements
We thank Ocean Adapt and NOAA-NEFSC for making data available from bottom trawl surveys and The Nature Conservancy for making substrate data available.

## Author contributions
S.M.R. designed research, conducted analysis, and wrote the manuscript with support from A.M.B. and P.N.H. A.M.B. conceived the original idea and P.N.H. supervised the project.

## Competing interests
The authors declare no competing interests.
