## [Peer Review File · Communications Biology]

Reviewers' comments:

Reviewer #1 (Remarks to the Author):

See attachment

Reviewer #2 (Remarks to the Author):

This paper investigates relationships between habitat variables and fish distribution shifts based on pelagic, demersal, and reef fish. They found that demersal fish are restricted by bottom type, pelagic fish by temperature, and reef fish by temperature and bottom type.

Overall, I think this paper was well written and organized and contributes a region-specific analysis of ongoing work understanding species-specific distribution shifts under climate change. I do think there are a few concepts to include or caveats to mention with this type of analysis.

It might be pertinent to include interaction variables (e.g., depth * bottom/surface temperature or substrate type * bottom temperature). I would expect that these variables are interacting and might explain even more of the demersal fish distribution deviance given that demersal fish may respond to bottom temperatures but are limited by bottom type in their habitat distribution. Fig 1+ 2 combined paired with the high deviance explained by depth, bottom temperature, and bottom salinity suggest to me that some interaction of these as a variable is worth considering... there is a literature in North Sea showing that demersal species go deeper (Climate change and deepening of the North Sea fish assemblage: a biotic indicator of warming seas. 2008. Dulvy et al). The percent deviance explained by bottom temperature (33%) for demersal species seems like a pretty large amount of deviance explained in ecology to be dismissed?

I like that the authors include an area occupied metric (overall range shift). Often it's not a matter of the center of the distribution but total range extent and leading/trailing edge of any observed expansion. You can see from some of your figures that the demersal fish range is potentially expanding, and it may be worth more detailed metrics of spatial distribution (e.g., Climate-Driven Shifts in Marine Species Ranges: Scaling from Organisms to Communities. 2020. Pinsky, Selden, Kitchen). There is also literature that is not cited in this paper suggesting that a simple area-weighted center of distribution can be biased (Model-based inference for estimating shifts in species distribution, area occupied and centre of gravity. 2016. Thorson, Pinsky, Ward) because of varying survey times/changes in distribution or timing of sampling effort that can alter at what point in seasonal migration we're observing centers of biomass, which has been shown in the NE US survey (Effect of spring onset and summer duration on fish species distribution and biomass along the Northeast United States Continental Shelf. 2017. Henderson, Mills, Thomas, Pershing, Nye). This caveat is worth mentioning in the discussion.

It's unclear to me why the first 5 years of data relative to the last 5 years of data are compared... why not look at the entire time series? We want to know if short term variability is different from longer time scale trends. This comparison of only 5 years may be just comparing short term variability in spatial distribution.

Are the percent deviance explained for the entire time series or only for a subset, such as the analysis for center of gravity? If only a subset, should be looked at for the entire time series to distinguish between long term trends and short term variability.

Line 20: "More associated with bottom substrate" than...? Temperature? Pelagic species? I think you mean compared to pelagic species but specify for clarity.

Reviewer #3 (Remarks to the Author):

The authors use a long-time series of trawl data to examine the role of general life history in species distribution changes. The authors use species distribution models to quantify factors important in determining species distributions: temperature, salinity, bottom habitat. They then compare the importance of these factors in the models to the general life history categories. They also compare the magnitude of changes in species distributions to the general life history categories. They find pelagic species moving more than reef species, moving more than demersal species. The "stickiness" of habitat is discussed.

The study is largely duplicative of an earlier study by McHenry et al but with a more specific focus on the relationship between general life history and species distribution changes. This focus means that the general life categorization becomes central to the results and I think the categories need to be revisited. As an example sand tiger sharks are associated with hard-bottom but can move hundreds of miles whereas spotfin butterflyfish are very site specific; both are categorized as reef fish. Similarly, the fringed filefish is categorized as reef but is frequently caught in the water column around Sargassum. Pearly razorfish, which inhabits sandy areas, are classified as reef fish. These are just some examples that may not affect the overall results, but since the main conclusions of the paper are contingent on these classifications, they are important. There are also species missing from their list: Atlantic cod, haddock, pollock. So the rationale for species selected is unclear.

The pelagic demersal split makes sense but it hard to define because of the continuum between them. The authors could include only those species that cleanly fit in the general categories.

The authors could also use a more granular classifications that distinguishes seasonal migrants from residents in addition to the pelagic - demersal categorization (see Able and Fahay 1998, 2010).

The authors could bypass the categorization of life history and compare the hypothesis of "stickiness" more directly. Species more dependent on temperature should move more. Species more dependent on habitat should move less. This would involve examining the correlation between percentage of deviance explained by temperature [habitat] and distribution metrics and testing for a positive [negative] correlation.

Some additional methodological details are needed. How did the authors combined trawl data across the change between vessels? Which seasonal survey is used (spring or fall)? How were missing environmental data were treated?

Review of Roberts et al

I am glad that someone has tackled this very important topic of research and I agree that the Northeast US is a good place to understand the relative importance of dynamic vs static habitat variables with its rich dataset and diverse species. The paper itself is very well written and the study is well-justified. However, I have major concerns about the methodology and cannot suggest it for publication in its current form. First, a common criticism of trawl surveys is that they poorly sample reef habitat and even areas with high rugosity. The net simply is ruined if you trawl in these areas so you likely do not sample reef fish and species that hang out in rocky substrate very well. Some justification on using a trawl survey to study reef fish must be given. Second, the species chosen seem very odd. Given that the NEFSC survey was designed to sample cod and haddock, why are these and the most common species in the survey not in the analysis? In looking at the list of species it seems like maybe the authors chose southern species, which I think is very admirable and interesting, but no justification is given. If looking at southern species is indeed a focus, please explain and shift the focus of the manuscript. It appears that the authors by choosing species for which there was >30% deviance explained inadvertently picked species that were poorly sampled and for which we can draw the least inference. Many of GAM models explain >90% of the variability for many species, suggesting extreme overfitting of the models. For example, in the case of twospot flounder 100% of the variability can be explained, but only 49 twospot flounder were caught over the entire time series. Very little inferences about habitat affinity can be made with 49 fish spread over 30 years and I would certainly not have confidence in a range shift calculated for such a small a sample size. For the period of 2013-2018 only 8 individuals for this species were caught. The high deviance explained for many species suggests a similar lack of generality for these species and I would not infer anything from them. Many of the species for which you have lower deviance explained have more observations. I think you need to pick some criteria where you throw out the species that aren't sampled frequently enough and keep the ones that do meet some threshold. Deviance explained for GAMs is not as useful as an assessment of skill done by reserving some of the data to test how well it predicts. This of course, is difficult or impossible to do with low sample size.

I'm not comfortable with calculating the shift in distribution by subtracting the mean center of biomass from 2 time stanzas, 1986-1991 and 2013-2018. The magnitude of the shift could very much be affected interannual spatio-temporal differences in sampling as well as climate variability (even though you mention that the NAO was not a significant explanatory variable in GAMs). The standard for reporting shifts in distribution is a linear regression over time although there is a paper that employed a differencing approach because of sample design (Walsh et al.) For some species zero species were caught in the latter part of the time series so it's unclear what the sample size is for all comparisons when a change in center of biomass cannot be calculated for some of the 191 species (i.e. hickory shad).

Methods

I wanted more information in the methods. Where were the data accessed? OceanAdapt only gives you mean latitude and longitude by species and year.

Please provide the strata that you used and then confirm that southern strata were sampled every year. While I don't think you have to necessarily use strata that were consistently sampled to develop the GAMs, if you are trying to make inferences about historical changes in center of biomass, area and min and max lat you do need to make sure you are using consistent strata. In Supp Fig 3 it appears that you used inshore strata which are not sampled after 2008. Very careful consideration of sampling design and consistency must be addressed.

Were calibrations between the Bigelow and Albatross around 2008 and 2009 employed? Please double check this and state how in methods. See the Miller et al. 2010 for details, but I would hope whoever gave you the data would have made the corrections.

Tow durations are not always 30 minutes, they can often be 20 minutes and I think after the switch to the Bigelow tows were limited to 20 minutes. How does that affect your GAMs? Are you using CPUE or absolute abundance from the tow?

Why did you choose to model just abundance in the GAMs and then used biomass-weighted center of biomass?

Please also give more details on the data from the Nature Conservancy. The weblink provided in the Data Availability section does not seem to work. I know for some of the Nature Conservancy's products they are not meant for use as site-specific information, but I am not familiar with all the benthic datasets out there. Is this grain size data and when was it collected?

Also, looking at the deviance explained for each individual variable it's clear that you have some correlated variables. Others have typically looked at VIF to eliminate some variables from the GAM. But since you aren't trying to come up with the "best" model perhaps use the correlation of variables to your advantage. For instance, in bank sea bass bottom type explains distribution much better than depth suggesting that it is truly the bottom type that they are selecting for. However, in many other species deviances explained are similar for depth and bottom type. I might suggest just getting rid of depth altogether, but perhaps there is a creative way to use correlated variables to your advantage to answer your research questions.

Sample sizes for all your comparisons and given in each figure would be useful and improve transparency.

Fig 3 caption reads that range is in meters squared, but title says km squared. I also was not sure what the numbers above the brackets meant.

A previous study highlighted the importance of a dynamic (temperature) and static (bottom rugosity) environmental variability in species habitat modeling and climate change projections. The dataset used therein might be of interest for this type of work.

I'm happy to provide assistance as you revise this study – Janet Nye

References

- Hare, J. A., J. Manderson, J. Nye, M. Alexander, P. Auster, D. Borggaard, A. Capotondi, K. Damon-Randall, E. Heupel, I. Mateo, L. O'Brien, D. Richardson, C. Stock, and S. T. Biegel. 2012. Cusk (*Brosme brosme*) and climate change: assessing the threat to a data poor candidate species under the Endangered Species Act. *ICES Journal of Marine Science* **69**:1753-1768.
- Miller, T. J., C. Das, P. J. Politis, A. S. Miller, S. M. Lucey, C. M. Legault, R. W. Brown, and P. J. Rago. 2010. Estimation of Albatross IV to Henry B. Bigelow calibration factors.
- Walsh, H. J., D. E. Richardson, K. E. Marancik, and J. A. Hare. 2015. Long-term changes in the distributions of larval and adult fish in the northeast US shelf ecosystem. *PLOS One* **10**.

Reviewer comments – Sticky Fish

Reviewer #1 (Remarks to the Author):

I am glad that someone has tackled this very important topic of research and I agree that the Northeast US is a good place to understand the relative importance of dynamic vs static habitat variables with its rich dataset and diverse species. The paper itself is very well written and the study is well-justified. However, I have major concerns about the methodology and cannot suggest it for publication in its current form. First, a common criticism of trawl surveys is that they poorly sample reef habitat and even areas with high rugosity. The net simply is ruined if you trawl in these areas so you likely do not sample reef fish and species that hang out in rocky substrate very well. Some justification on using a trawl survey to study reef fish must be given. **Note, we have redone the analysis with species sampled consistently every year. This removed all but one reef species. Thus, we have changed the species type classifications to be benthic (species that inhabit the seafloor such as flounder and plaice), demersal (species that inhabit near the seafloor) and pelagic (species that inhabit the water column) based on fishbase classifications.**

Second, the species chosen seem very odd. Given that the NEFSC survey was designed to sample cod and haddock, why are these and the most common species in the survey not in the analysis? In looking at the list of species it seems like maybe the authors chose southern species, which I think is very admirable and interesting, but no justification is given. If looking at southern species is indeed a focus, please explain and shift the focus of the manuscript. It appears that the authors by choosing species for which there was >30% deviance explained inadvertently picked species that were poorly sampled and for which we can draw the least inference. Many of GAM models explain >90% of the variability for many species, suggesting extreme overfitting of the models. For example, in the case of twospot flounder 100% of the variability can be explained, but only 49 twospot flounder were caught over the entire time series. Very little inferences about habitat affinity can be made with 49 fish spread over 30 years and I would certainly not have confidence in a range shift calculated for such a small a sample size. For the period of 2013-2018 only 8 individuals for this species were caught. The high deviance explained for many species suggests a similar lack of generality for these species and I would not infer anything from them. Many of the species for which you have lower deviance explained have more observations. I think you need to pick some criteria where you throw out the species that aren't sampled frequently enough and keep the ones that do meet some threshold. Deviance explained for GAMs is not as useful as an assessment of skill done by reserving some of the data to test how well it predicts. This of course, is difficult or impossible to do with low sample size. **We have redone the analysis to include 114 species that were collected in 17 out of the 35 years in both the spring and the fall. A list of the species abundances per year can be found in supplemental materials table 5.**

I'm not comfortable with calculating the shift in distribution by subtracting the mean center of biomass from 2 time stanzas, 1986-1991 and 2013-2018. The magnitude of the shift could very much be affected interannual spatio-temporal differences in sampling as well as climate variability (even though you mention that the NAO was not a significant explanatory variable in

GAMs). The standard for reporting shifts in distribution is a linear regression over time although there is a paper that employed a differencing approach because of sample design (Walsh et al.) For some species zero species were caught in the latter part of the time series so it's unclear what the sample size is for all comparisons when a change in center of biomass cannot be calculated for some of the 191 species (i.e. hickory shad).

We only calculated changes in the center of biomass for species that were collected in both time periods. We chose to use five-year time chunks because this period of time should account for variability that may be experienced by the NAO. This way, we can boil down the distance shifted to one number over the entire time period in order to compare that to the most important predictor variable for these species (and the species-types). If we did a linear regression we would not be able to make these comparisons.

Methods I wanted more information in the methods. Where were the data accessed? OceanAdapt only gives you mean latitude and longitude by species and year.

We accessed the data from OceanAdapt. In ocean adapt you can download the latitude and longitude by year as well as the raw data (that has been calibrated between boats).

Please provide the strata that you used and then confirm that southern strata were sampled every year. While I don't think you have to necessarily use strata that were consistently sampled to develop the GAMs, if you are trying to make inferences about historical changes in center of biomass, area and min and max lat you do need to make sure you are using consistent strata. In Supp Fig 3 it appears that you used inshore strata which are not sampled after 2008. Very careful consideration of sampling design and consistency must be addressed.

The fall stratum used were

1010 1020 1030 1040 1050 1060 1070 1080 1090 1100 1110 1120 1130 1140 1150 1160 1170
1180 1190 1200 1210 1220 1230 1240 1250 1260 1270 1280 1290 1340 1360 1370 1380 1390
1400 1610 1620 1630 1650 1660 1670 1690 1700 1710 1730 1740 1750 1760 3020 3050 3080
3110 3140 3170 3200 3230 3260 3290 3320 3350 3380 3410 3440 3450 3460 3600 3610

And the spring stratum used were

[1] 1010 1020 1030 1040 1050 1060 1070 1080 1090 1100 1110 1130 1140 1150 1160 1170
1180 1190 1200 1210 1220 1230 1240 1250 1260 1270 1280 1290 1300 1340 1360 1370 1380
1390 1400 1610 1620 1650 1660 1670 1680 1690 1700 1710 1730 1740 1750 3020 3050 3080
3110 3140 3170 3200 3230 3260 3290 3320 3350 3380 3410 3440 3450 3460 3590 3600 3610
3660

We've updated the map to depict this (supplemental Figure S5). These stratum were sampled every year (from 1985-2018).

Were calibrations between the Bigelow and Albatross around 2008 and 2009 employed? Please double check this and state how in methods. See the Miller et al. 2010 for details, but I would hope whoever gave you the data would have made the corrections.

The Ocean Adapt data has been calibrated between the Bigelow and Albatross. I checked with one of the post-docs in Malin Pinsky's lab to make sure this is the case.

Tow durations are not always 30 minutes, they can often be 20 minutes and I think after the switch to the Bigelow tows were limited to 20 minutes. How does that affect your GAMs? Are you using CPUE or absolute abundance from the tow?

We made sure to use Ocean Adapts CPUE this time.

Why did you choose to model just abundance in the GAMs and then used biomass-weighted center of biomass?

See comment above.

Please also give more details on the data from the Nature Conservancy. The weblink provided in the Data Availability section does not seem to work. I know for some of the Nature Conservancy's products they are not meant for use as site-specific information, but I am not familiar with all the benthic datasets out there. Is this grain size data and when was it collected?

The link takes you to the NAMERA assessment where you can read the chapters about the study (with more information on how the data were collected) and download the spatial data (by following the download spatial data tab). We used the benthic habitats data which is free to download. The sediment data we used was grain size in mm which has been added to the caption for Supplemental Figure S5 and line 219 in the methods. Sediment data were collected between 1881 and 1992

<http://www.conservationgateway.org/ConservationByGeography/NorthAmerica/UnitedStates/edc/Documents/Chapter-3-Benthic-Habitatas-20100329.pdf>

Also, looking at the deviance explained for each individual variable it's clear that you have some correlated variables. Others have typically looked at VIF to eliminate some variables from the GAM. But since you aren't trying to come up with the "best" model perhaps use the correlation of variables to your advantage. For instance, in bank sea bass bottom type explains distribution much better than depth suggesting that it is truly the bottom type that they are selecting for. However, in many other species deviances explained are similar for depth and bottom type. I might suggest just getting rid of depth altogether, but perhaps there is a creative way to use correlated variables to your advantage to answer your research questions.

We added table 6 in the supplemental figures to examine correlation across variables. It looks like bottom temperature and bottom salinity are correlated in the spring (.79), but we do not believe this is an issue as we are mostly reporting on the fall results.

Sample sizes for all your comparisons and given in each figure would be useful and improve transparency.

We have added this to the figures.

Fig 3 caption reads that range is in meters squared, but title says km squared. I also was not sure what the numbers above the brackets meant.

We've changed this calculation to be percentage change in range size (to account for species that have large ranges and have shifted a little which could have come up as a large change in range size, but not when you look at percentage change).

A previous study highlighted the importance of a dynamic (temperature) and static (bottom rugosity) environmental variability in species habitat modeling and climate change projections. The dataset used therein might be of interest for this type of work.

I'm happy to provide assistance as you revise this study – Janet Nye

Thank you for all of the helpful suggestions!

References Hare, J. A., J. Manderson, J. Nye, M. Alexander, P. Auster, D. Borggaard, A. Capotondi, K. Damon-Randall, E. Heupel, I. Mateo, L. O'Brien, D. Richardson, C. Stock, and S. T. Biegel. 2012. Cusk (*Brosme brosme*) and climate change: assessing the threat to a data poor candidate species under the Endangered Species Act. *ICES Journal of Marine Science* 69:1753-1768. Miller, T. J., C. Das, P. J. Politis, A. S. Miller, S. M. Lucey, C. M. Legault, R. W. Brown, and P. J. Rago. 2010. Estimation of Albatross IV to Henry B. Bigelow calibration factors. Walsh, H. J., D. E. Richardson, K. E. Marancik, and J. A. Hare. 2015. Long-term changes in the distributions of larval and adult fish in the northeast US shelf ecosystem. *PLOS One* 10.

Reviewer #2 (Remarks to the Author):

This paper investigates relationships between habitat variables and fish distribution shifts based on pelagic, demersal, and reef fish. They found that demersal fish are restricted by bottom type, pelagic fish by temperature, and reef fish by temperature and bottom type.

Overall, I think this paper was well written and organized and contributes a region-specific analysis of ongoing work understanding species-specific distribution shifts under climate change. I do think there are a few concepts to include or caveats to mention with this type of analysis.

It might be pertinent to include interaction variables (e.g., depth * bottom/surface temperature or substrate type * bottom temperature). I would expect that these variables are interacting and might explain even more of the demersal fish distribution deviance given that demersal fish may respond to bottom temperatures but are limited by bottom type in their habitat distribution. Fig 1+ 2 combined paired with the high deviance explained by depth, bottom temperature, and bottom salinity suggest to me that some interaction of these as a variable is worth considering... there is a literature in North Sea showing that demersal species go deeper (Climate change and deepening of the North Sea fish assemblage: a biotic indicator of warming seas. 2008. Dulvy et al). The percent deviance explained by bottom temperature (33%) for demersal species seems like a pretty large amount of deviance explained in ecology to be dismissed?

We have redone our analysis to incorporate only demersal, pelagic and benthic species. After doing this we see that demersal species are explained by bottom temperature and substrate, and we have shifted our focus to look at historic distribution shifts versus the strongest predictor variable for all species. We believe that this has improved the argument and lessened the necessity of the pelagic vs. demersal split. Additionally, we include a discussion on the complex results for demersal species that may make them more difficult to predict under climate change and incorporated the above mentioned reference (line 165).

I like that the authors include an area occupied metric (overall range shift). Often it's not

a matter of the center of the distribution but total range extent and leading/trailing edge of any observed expansion. You can see from some of your figures that the demersal fish range is potentially expanding, and it may be worth more detailed metrics of spatial distribution (e.g., Climate-Driven Shifts in Marine Species Ranges: Scaling from Organisms to Communities. 2020. Pinsky, Selden, Kitchen). There is also literature that is not cited in this paper suggesting that a simple area-weighted center of distribution can be biased (Model-based inference for estimating shifts in species distribution, area occupied and centre of gravity. 2016. Thorson, Pinsky, Ward) because of varying survey times/changes in distribution or timing of sampling effort that can alter at what point in seasonal migration we're observing centers of biomass, which has been shown in the NE US survey (Effect of spring onset and summer duration on fish species distribution and biomass along the Northeast United States Continental Shelf. 2017. Henderson, Mills, Thomas, Pershing, Nye). This caveat is worth mentioning in the discussion.

We redid this analysis to only use strata that were sampled every year and to split up the analysis into spring and fall surveys. We hope that this has addressed the above issue. We added the above-mentioned caveat in the discussion section (lines 185-195) as well as references to the literature mentioned.

It's unclear to me why the first 5 years of data relative to the last 5 years of data are compared... why not look at the entire time series? We want to know if short term variability is different from longer time scale trends. This comparison of only 5 years may be just comparing short term variability in spatial distribution.

We only calculated changes in the center of biomass for species that were collected in both time periods. We chose to use five-year time chunks because this period of time should account for variability that may be experienced by the NAO. This way, we can boil down the distance shifted to one number over the entire time period in order to compare that to the most important predictor variable for these species (and the species-types). If we did a linear regression we would not be able to make these comparisons.

Are the percent deviance explained for the entire time series or only for a subset, such as the analysis for center of gravity? If only a subset, should be looked at for the entire time series to distinguish between long term trends and short term variability.

Percent deviance explained is for the entire time series. We added this in figure caption for figure 2 (line 112) and line 231.

Line 20: "More associated with bottom substrate" than...? Temperature? Pelagic species? I think you mean compared to pelagic species but specify for clarity.
added than other dynamic environmental variables

Reviewer #3 (Remarks to the Author):

The authors use a long-time series of trawl data to examine the role of general life history in species distribution changes. The authors use species distribution models to quantify factors important in determining species distributions: temperature, salinity, bottom habitat. They then compare the importance of these factors in the models to the general life history categories. They also compare the magnitude of changes in species distributions to the general life history categories. They find pelagic species moving more than reef species, moving more than demersal species. The "stickiness" of habitat is discussed.

The study is largely duplicative of an earlier study by McHenry et al but with a more specific focus on the relationship between general life history and species distribution changes. This focus means that the general life categorization becomes central to the results and I think the categories need to be revisited. As an example sand tiger sharks are associated with hard-bottom but can move hundreds of miles whereas spotfin butterflyfish are very site specific; both are categorized as reef fish. Similarly, the fringed filefish is categorized as reef but is frequently caught in the water column around Sargassum. Pearly razorfish, which inhabits sandy areas, are classified as reef fish. These are just some examples that may not affect the overall results, but since the main conclusions of the paper are contingent on these classifications, they are important. There are also species missing from their list: Atlantic cod, haddock, pollock. So the rationale for species selected is unclear.

Note, we have redone the analysis with species sampled consistently every year. This removed all but one reef species. Thus, we have changed the species type classifications to be benthic

(species that inhabit the seafloor such as flounder and plaice), demersal (species that inhabit near the seafloor) and pelagic (species that inhabit the water column) based on fishbase classifications.

The pelagic demersal split makes sense but it hard to define because of the continuum between them. The authors could include only those species that cleanly fit in the general categories.

see comment above. Hopefully this split is more specific now that we have removed certain species.

The authors could also use a more granular classifications that distinguishes seasonal migrants from residents in addition to the pelagic - demersal categorization (see Able and Fahay 1998, 2010).

Note, we have redone the analysis with species sampled consistently every year. This removed all but one reef species. Thus, we have changed the species type classifications to be benthic (species that inhabit the seafloor such as flounder and plaice), demersal (species that inhabit near the seafloor) and pelagic (species that inhabit the water column) based on fishbase classifications.

The authors could bypass the categorization of life history and compare the hypothesis of "stickiness" more directly. Species more dependent on temperature should move more. Species more dependent on habitat should move less. This would involve examining the correlation between percentage of deviance explained by temperature [habitat] and distribution metrics and testing for a positive [negative] correlation.

We have done this now by including figure 1 which shows historic distribution shifts versus the strongest predictor variable for this species. We believe that this has improved the argument, and lessened the necessity of the pelagic vs. demersal split. We are greatly appreciative of this feedback.

Some additional methodological details are needed. How did the authors combined trawl data across the change between vessels? Which seasonal survey is used (spring or fall)? How were missing environmental data were treated?

The Ocean Adapt data has been calibrated between the Bigelow and Albatross. I checked with one of the post-docs in Malin Pinsky's lab to make sure this is the case. We specify this in the methods (line 204). We split the study into spring and fall and mostly report on fall results (however spring is in the supplemental materials). We only used in situ data BT and salinity and there was very little missing data for depth and substrate. We removed missing values in the analysis for each GAM (line 226).

Reviewers' comments:

Reviewer #1 (Remarks to the Author):

The authors have responded to all my concerns admirably. I have only a few minor concerns:

1-I think the brackets at the top of all your figures is the p value. You don't specify in any of the figure captions. If you keep the brackets I would specify what it is. However, I would suggest not keeping them because they take up half of your figure space and so it's hard to see any differences in the box plots.

2-The formatting of the supplementary material is wonky with figure captions not on the same page. Editorial staff tend not to look at Supp Material so please take it upon yourself to make it clear to readers. I would suggest a separate page for each figure so the formatting doesn't get wonky. These days some of the most important material is in the supplemental material

3-Fig s5-occurrences are much more relevant than abundance esp for those pelagic species where you find them only at a few stations. I'd put number of stations found total or even by the 5 yr time blocks since that is the criteria you used to include them

4-Line 36, you should also cite Pinsky et al. 2013 Science paper. That correlates shifts to temperature better than Nye et al.

5-Why do you think the shifts in relation to bottom salinity are more evident in the Fall?

Reviewer #2 (Remarks to the Author):

I'm still uncomfortable with the split of biomass into 2 time periods, 1986-1991 and 2013-2018, to understand the shift in distribution for the GAM. These shorter time periods leave open the opportunity to be interpreting short term spatiotemporal differences rather than shifts in distribution, as well as patterns in climate variability. The authors did present their argument for why they did it this way, that the time-period over which the NAO would vary is 5 years. I would still present a GAM with the data standardized relative to the entire time series, because a 5 year time period analysis can simply be identifying small differences in short-term spatial distribution of a species due to climate variability and other environmental & biological factors, however I leave this up to the editor.

Other than that, the authors addressed all comments and you can tell put a lot of work into these revisions to address all reviewers concerns, nice work!

Reviewer #3 (Remarks to the Author):

The authors have well addressed the comments of the reviewers and in my opinion, the manuscript is much improved. I have three comments, which are similar to previous comments that require additional revision.

First, I believe the authors are overselling the novelty of their work. Benthic habitat variables have been included in species distribution models in the Northeast region using the same data as used here for almost 10 years (for example Hare et al. 2012, Morley et al. 2018, McHenry et al. 2019). The results reported in the manuscript confirm and build upon these early studies. This is by no means the first study to include benthic habitat characteristics in projections; this study does add the examination of the relationship between broad life history characteristics and projected distribution change (pelagic, demersal, benthic).

Second, the categorization of species still needs work. Looking through the list: *Trachurus lathmi* was listed as a demersal species (I would classify pelagic), *Ophidion marginatum* was listed a demersal (I would classify benthic - a burrower), *Centropristis striata* was listed as pelagic (I would classify demersal). I recommend that the authors have a fish ecologist or ichthyologist review their classifications - a majority are correct but at least three a questionable. This is only going to have a minor impact on the analyses if any, but I feel it is important to be a correct as possible.

Third, in the Supplemental Materials, Genus should be capitalized for the scientific names.

Response to Reviewers

Reviewer #1 (Remarks to the Author):

The authors have responded to all my concerns admirably. I have only a few minor concerns:

1-I think the brackets at the top of all your figures is the p value. You don't specify in any of the figure captions. If you keep the brackets I would specify what it is. However, I would suggest not keeping them because they take up half of your figure space and so it's hard to see any differences in the box plots.

Thank you for this comment. We have added a short description in the captions for Figs. 1 and 3 (and supplemental Figs 1, 3, 4).

2-The formatting of the supplementary material is wonky with figure captions not on the same page. Editorial staff tend not to look at Supp Material so please take it upon yourself to make it clear to readers. I would suggest a separate page for each figure so the formatting doesn't get wonky. These days some of the most important material is in the supplemental material

We have adjusted the supplemental materials so that each figure is on a separate page.

3-Fig s5-occurrences are much more relevant than abundance esp for those pelagic species where you find them only at a few stations. I'd put number of stations found total or even by the 5 yr time blocks since that is the criteria you used to include them

We have added two columns to Supplemental tables S5 and S6. These columns address this issue by noting the fall and spring abundance in the first and last five year periods.

4-Line 36, you should also cite Pinsky et al. 2013 Science paper. That correlates shifts to temperature better than Nye et al.

We have added this citation to line 36.

5-Why do you think the shifts in relation to bottom salinity are more evident in the Fall?

The stronger influence of salinity on distribution shifts in the fall is likely due to the combined effect of a drier and warmer weather that has led to more drastic changes in ocean salinity in the fall (less precipitation and warmer weather leads to more evaporation) which most likely effects species distributions more than in the cooler and wetter spring. We are happy to add this discussion into supplemental materials as we do not know if it completely fits within the short format paper constraints.

Reviewer #2 (Remarks to the Author):

I'm still uncomfortable with the split of biomass into 2 time periods, 1986-1991 and 2013-2018, to understand the shift in distribution for the GAM. These shorter time periods leave open the opportunity to be interpreting short term spatiotemporal differences rather than

shifts in distribution, as well as patterns in climate variability. The authors did present their argument for why they did it this way, that the time-period over which the NAO would vary is 5 years. I would still present a GAM with the data standardized relative to the entire time series, because a 5 year time period analysis can simply be identifying small differences in short-term spatial distribution of a species due to climate variability and other environmental & biological factors, however I leave this up to the editor.

We have addressed these concerns through clarification with the editor and reviewer. We have added a better description of this process in the methods and body of the paper. In line 72 we added "between 1985-2018" to note that the GAMs were done for the entire time period, and in lines 76-77 we note that the geographic distributions were calculated for 5 year periods. We added similar clarifications in the methods (lines 264 and 269).

Other than that, the authors addressed all comments and you can tell put a lot of work into these revisions to address all reviewers concerns, nice work!

Reviewer #3 (Remarks to the Author):

The authors have well addressed the comments of the reviewers and in my opinion, the manuscript is much improved. I have three comments, which are similar to previous comments that require additional revision.

First, I believe the authors are overselling the novelty of their work. Benthic habitat variables have been included in species distribution models in the Northeast region using the same data as used here for almost 10 years (for example Hare et al. 2012, Morley et al. 2018, McHenry et al. 2019). The results reported in the manuscript confirm and build upon these early studies. This is by no means the first study to include benthic habitat characteristics in projections; this study does add the examination of the relationship between broad life history characteristics and projected distribution change (pelagic, demersal, benthic).

Thank you for this comment. We hope that our discussion section addresses the fact that we are building on this work. Lines 160-161 outline how we are building on work by McHenry et al 2019 and we added lines 161-163 to tie this to Morely et al which used sediment in their future projections. We have added line 178-180 which addresses the Hare et al 2018 study you have suggested.

Second, the categorization of species still needs work. Looking through the list: *Trachurus lathmi* was listed as a demersal species (I would classify pelagic), *Ophidion marginatum* was listed a demersal (I would classify benthic - a burrower), *Centropristis striata* was listed as pelagic (I would classify demersal). I recommend that the authors have a fish ecologist or ichthyologist review their classifications - a majority are correct but at least three a

questionable. This is only going to have a minor impact on the analyses if any, but I feel it is important to be as correct as possible.

We agree that these three species were misclassified. We went through an additional extensive review of the studied species (compared to McHenry et al) and reclassified the three species as noted. We had mistakenly classified the three species and when we looked back at McHenry et al's study, they also had them classified as you suggested. We apologize for this mistake. These new classifications did not change the study results.

Third, in the Supplemental Materials, Genus should be capitalized for the scientific names. We have corrected this in the supplemental tables.

REVIEWERS' COMMENTS:

Reviewer #2 (Remarks to the Author):

Nice work! Authors have addressed all previous reviews & comments thoroughly & adequately.

Reviewer #3 (Remarks to the Author):

The authors have adequately addressed my comments. The revised version is much improved.